# Beyond MicroRNAs: Emerging Role of Other Non-Coding RNAs in HPV-Driven Cancers

**DOI:** 10.3390/cancers12051246

**Published:** 2020-05-15

**Authors:** Mariateresa Casarotto, Giuseppe Fanetti, Roberto Guerrieri, Elisa Palazzari, Valentina Lupato, Agostino Steffan, Jerry Polesel, Paolo Boscolo-Rizzo, Elisabetta Fratta

**Affiliations:** 1Division of Immunopathology and Cancer Biomarkers, Centro di Riferimento Oncologico di Aviano (CRO), IRCCS, National Cancer Institute, 33081 Aviano (PN), Italy; mtcasarotto@cro.it (M.C.); roberto.guerrieri@cro.it (R.G.); asteffan@cro.it (A.S.); 2Division of Radiotherapy, Centro di Riferimento Oncologico di Aviano (CRO), IRCCS, National Cancer Institute, 33081 Aviano (PN), Italy; giuseppe.fanetti@cro.it (G.F.); elisa.palazzari@cro.it (E.P.); 3Division of Otolaryngology, General Hospital “Santa Maria degli Angeli”, 33170 Pordenone, Italy; valentinalupato@gmail.com; 4Division of Cancer Epidemiology, Centro di Riferimento Oncologico di Aviano (CRO), IRCCS, National Cancer Institute, 33081 Aviano (PN), Italy; polesel@cro.it; 5Section of Otolaryngology, Department of Neurosciences, University of Padova, 31100 Treviso, Italy; paolo.boscolorizzo@unipd.it

**Keywords:** HPV, squamous cell carcinoma, non-coding RNAs, circular RNAs, PIWI-interacting RNAs, long non-coding RNAs

## Abstract

Persistent infection with high-risk Human Papilloma Virus (HPV) leads to the development of several tumors, including cervical, oropharyngeal, and anogenital squamous cell carcinoma. In the last years, the use of high-throughput sequencing technologies has revealed a number of non-coding RNA (ncRNAs), distinct from micro RNAs (miRNAs), that are deregulated in HPV-driven cancers, thus suggesting that HPV infection may affect their expression. However, since the knowledge of ncRNAs is still limited, a better understanding of ncRNAs biology, biogenesis, and function may be challenging for improving the diagnosis of HPV infection or progression, and for monitoring the response to therapy of patients affected by HPV-driven tumors. In addition, to establish a ncRNAs expression profile may be instrumental for developing more effective therapeutic strategies for the treatment of HPV-associated lesions and cancers. Therefore, this review will address novel classes of ncRNAs that have recently started to draw increasing attention in HPV-driven tumors, with a particular focus on ncRNAs that have been identified as a direct target of HPV oncoproteins.

## 1. Introduction

Worldwide, 4.5% of all cancers (630,000 new cancer cases per year) are attributable to Human Papilloma Virus (HPV) infection [1]. HPVs are a heterogeneous group of small non-envelope double-stranded circular DNA viruses targeting the basal cells of stratified epithelia [2,3]. The IARC Working Group has classified alpha-HPV types 16, 18, 31, 33, 35, 39, 45, 51, 52, 56, 58, 59 as carcinogenic to humans; these high-risk (HR)-HPVs are responsible for virtually all carcinomas of the cervix and different proportions of carcinomas of the anus, vagina, penis, vulva, and oropharynx (Table 1) [4]. Among the HR-HPV types, HPV16 is responsible for the majority of HPV-driven cancers. In addition, some HPV types of the beta genus showing cutaneous tropism have been proposed to cooperate with ultraviolet radiation in the development of non-melanoma skin cancer [5].

Cervical squamous cell carcinoma (CSCC) is the fourth most common cancer in women worldwide [16]. Despite the spread of screening programs has significantly reduced mortality, nearly 50% of patients worldwide are still diagnosed with locally advanced stages. Concurrent platinum based chemoradiation is the current standard treatment of locally advanced CSCC [17]. Several studies have shown improved local control and survival with the use of concurrent chemoradiation with respect to radiotherapy alone but in these patients, recurrence rate and mortality remain still high [18,19]. Infection with HR-HPV is the most significant risk factor for CSCC. Several studies shown that the sustained expression of the oncogenic genes E6 and E7 of HPV is involved in CSCC progression [20,21,22,23,24] but the prognostic role of HPV expression genes is not fully elucidated yet. In clinical practice there are not available prognostic factors that can guide therapeutic choice in CSCC patients, and several studies are needed to improve our knowledge, especially on the role of HPV and other molecular and genomic factors.

The role of HPV in head and neck squamous cell carcinoma (HNSCC) has emerged in the last decades, with relevant etiological and clinical aspects. Nowadays, approximately 30% of oropharyngeal squamous cell carcinoma (OPSCC) is attributable to HPV worldwide [1], but this proportion is expected to increase in the close future. Therefore, HPV has been included as one of the strongest prognostic factors of OPSCC alongside the already well-defined stage, smoking, performance status, and quality of treating facilities [25]. Compared to HPV-negative counterparts, HPV-positive OPSCC patients show peculiar clinico-pathological features and improved prognosis [26]. On this basis, a different TNM staging has been proposed for HPV-positive OPSCC [27]. Notably, a gender-specific trend has also emerged for HPV-driven OPSCC. In fact, mirroring the downward trend of CSCC due to HPV vaccination programs, the HPV-driven OPSCC incidence is expected to decline in women, whereas the incidence among men has been increasing over the last years [28]. One possible explanation could lie in the profound differences observed in male versus female immune responses in cancer since it has become increasingly evident that the major susceptibility of women to a variety of autoimmune diseases might contribute to enhanced immune surveillance against various tumor types [29]. Sex hormones can also affect the immune system since high estrogen levels have been shown to promote antibody production, whereas androgens have been reported to suppress immune function [30]. Consistent with this evidence, only a small proportion of seroconversions occur in men following HPV infection [31], and HPV seroprevalence in men is significantly lower than that reported among women [32]. The combination of clinical stage, HPV status, and smoking history lead to the definition of three different OPSCC risk groups with different prognosis [33]. Despite a more precise risk assessment, the therapeutic options remain unchanged and include chemo-radiation or surgery with or without adjuvant (chemo-) radiotherapy in the radical setting [34]. Due to the evidence of a better prognosis in HPV-driven OPSCC, several strategies in treatment de-intensification are under evaluation with the purpose to maintain efficacy and reduce short- and long-term treatment related side effects [35,36,37,38,39,40,41,42]. These studies are only part of a growing literature in the field of reduction of aggressiveness of treatments for HPV-driven OPSCC, and mainly focus on the low risk OPSCC patients. Hopefully, other trials that are still ongoing may also help clinicians in the choice of the optimal strategy to offer to HPV-driven OPSCC (in particular results from PATHOS, HN002, HN005, and KEYCHAIN trials). Although clinical trials move in the direction of reduced treatment intensity, 20% of HPV-positive OPSCC patients relapse and even die of the disease [43]. For that reason, the identification of novel prognostic factors is urgently needed not only to select HPV-positive OPSCC patients that may benefit from de-intensified treatments, but also to identify those patients at higher risk of relapse.

Squamous cell carcinoma of the anal canal (SCCA) is a rare cancer [44] associated with HPV infection in 80–85% of patients (usually HPV16 or HPV18 genotypes in Europe) [45]. Other important risk factors for SCCA include human immunodeficiency virus, immune suppression in transplant recipients, and the use of immunosuppressant drugs. Definitive chemoradiotherapy (CRT) is the current standard of care for patients with locally advanced SCCA, whereas surgery as a salvage treatment is indicated for patients with persistent disease after CRT or local relapse [46]. Prognostic factors for survival in SCCA include male sex, positive lymph nodes, and primary tumor size greater than 5 cm [46]. Despite advances in the understanding of biology and pathogenesis of SCCA, there is considerable heterogeneity in terms of outcome, particularly for more advanced stages. Only a limited number of biomarkers have been investigated and at present there are not current available factors to guide prognosis or select treatment. In a systematic review, Lampejo et al. [47] examined 29 different biomarkers, but the tumor suppressor genes p53 and p21 were the only significantly related to prognosis. Therefore, since there are no current biomarkers that strongly predict response to CRT and prognosis in SCCA patients, the investigation of HPV-related biomarkers would be an interesting objective.

## 2. HPV-Driven Cancerogenesis

The HPV genome is organized into three regions: a non-coding region, termed the long control region, which contains the early promoter and regulatory element involved in viral replication and transcription, and two protein-coding regions, the early (E) region coding proteins regulating viral transcription (E2), viral DNA replication (E1, E2), cell proliferation (E5, E6, E7), and viral particle release (E4), and the late (L) region which encodes the structural proteins (L1 and L2). E5, E6, and E7 are viral oncogenes; several studies on mucosal HR HPVs have established that E6 and E7 play a pivotal role in altering host immune response and promoting cell proliferation and transformation [3].

The best characterized interactions, whose maintenance is considered fundamental for the neoplastic phenotype, are those between E6 and E7 with p53 and pRb, respectively. By suppressing p53 activity, HPV is able to bypass cellular senescence. On the other hand, the release of E2F transcription factors allows for unscheduled cell proliferation [48]. The E6 oncoprotein of HR-HPV binds the E6-associate protein (E6AP), an E3 ubiquitin ligase that ubiquitinates target proteins for subsequent proteasome degradation. P53, a transcription factor that induces cell cycle arrest or apoptosis in response to cellular stress or DNA damage, is the best characterized target of E6/E6AP heterodimer-induced degradation leading to the loss of tumor suppression activity, accumulation of DNA mutations, and to genomic instability [49]. E7 of HR-HPV types interacts and inactivates pRb and related pocket proteins (p107 and p130), which is in control of the G1-S phase transition by binding the transcription factor E2F [50]. As a consequence, E2F is released, with consequent promotion of cell G1-S phase transition, and transcription of genes, such as cyclin E and cyclin A, which are required for cell cycle progression [51]. Furthermore, by recruiting p300/CBP and pRb, E7 brings the histone acetyltransferase domain of p300/CBP into proximity to pRb and promotes its acetylation, leading to cell cycle deregulation [52]. In addition, cells harboring transforming HR-HPV infection acquire the capability to replicate indefinitely through the ability of E6 to reactivate the expression of telomerase (a ribonucleoprotein complex containing an internal RNA component and a catalytic protein, TERT, with telomere specific reverse transcriptase activity) by significantly upregulating TERT promoter activity [53].

Although the main mechanism of the malignant transformation induced by HPV is orchestrated by the abovementioned transforming activity of the viral E6 and E7 oncoproteins, the control of gene expression by specific non-coding RNAs (ncRNAs) may give a significant contribution in the process of transformation. Regarding the relationship between transforming HPV infection and the expression pattern of host ncRNAs, numerous studies, mainly focused on the study of micro RNAs (miRNAs), have shown a different miRNAs expression in HPV-positive tumor cells compared to the negative counterpart [54,55,56]. However, different investigations give conflicting results with a significant proportion of miRNAs being upregulated in one study but downregulated in another study [57]. Interestingly, both by standard sequencing and next generation sequencing, it has been successfully demonstrated that HPVs are able to generate both their own miRNAs and circular RNAs (circRNAs) [58,59]. Although the levels of expression are rather low, the frequent identification of viral miRNAs in cell lines and their higher expression in high-grade lesions suggest that they probably have a role in viral replication and malignant transformation [58].

Overall, although the transforming activity of HPV is mainly based on the degradation of p53 and pRb induced, respectively, by the viral oncoproteins E6 and E7, numerous other mechanisms including the contribution of ncRNAs generated both by the host cell and by the virus seem to participate in the process of carcinogenesis and tumor progression of HPV-induced tumors.

## 3. Non-Coding RNAs

In the last years, ncRNAs have emerged as key players in regulating the expression levels of the coding RNAs and other cellular processes [60]. Generally, ncRNAs with lengths exceeding 200 nucleotides are known as long non-coding RNAs (lncRNAs) or circRNAs, whereas all smaller transcripts are defined as small ncRNAs (sncRNAs); among sncRNAs small interfering RNAs (siRNAs), miRNAs, and P-element-induced wimpy testis (PIWI)-interacting RNAs (piRNAs) have been extensively studied so far (Figure 1) [61]. With the development of high-throughput sequencing technology and bioinformatics, an increasing number of ncRNAs are gradually being discovered. To date, multiple functional tumor-associated ncRNAs have been described, and several studies have shown they have either oncogenic or tumor-suppressive properties in cancer (for review see Diamantopoulos et al. [62]). Increasing evidence has revealed that ncRNAs play key roles not only in tumor progression and metastasis, but also in chemoresistance [63,64,65,66]. In particular, ncRNAs have been found to act as mediators of drug-resistance mechanisms through their ability to impair cell cycle arrest and apoptosis [67], but also to induce and modulate epithelial–mesenchymal transition (EMT) and cell adhesion-associated signaling pathways [68,69,70].

Besides miRNAs, that have been extensively studied in the last years [20,24,71], other ncRNAs (i.e., circRNAs, piRNAs, and lncRNAs) are drawing increasing attention nowadays since they have been found to play a role in HPV-driven tumors, suggesting that they could function as predictive biomarkers and therapeutic targets. In fact, many circRNAs, piRNAs, and lncRNAs involved in HPV-driven tumors have recently been characterized and several models of action have also been proposed; in some cases, deregulation of a specific ncRNA has come from two or more different studies (Table A1). Given these considerations, in this review we mainly focus our attention on these ncRNAs, classifying them according to the different HPV-driven tumor types.

## 4. CircRNAs Expression in HPV-Driven Cancers

Even though circRNAs are derived from linear pre-mRNAs, they are generally presented as covalently linked circles lacking both 5′ cap and 3′ poly(A) tails [72]. Over 80% of the identified circRNAs is exon-derived circRNA, containing only exons and completely lacking introns [73]. However, the splicing mechanism of circRNAs is complicated and the same position of a gene can produce different types of circRNAs. Consistently, three other types of circRNAs have been also identified by high-throughput sequencing: circular intronic RNAs, which consists of only introns, exon-intron cirRNAs, in which one intron is inserted between two exons, and tRNA intronic circRNAs, which can form stable circRNA via pre-tRNA splicing [73]. One of the most widely studied functions of circRNAs is their role as miRNA sponges and as modulators of splicing or transcription. In addition, circRNAs interact with RNA-binding proteins, and transport substances and information. Furthermore, the presence of short sequences containing N6-methyladenosine (m6A) as internal ribosomal entry site [74] allows to a small number of circRNAs to be translated into peptides or proteins that are functionally different from their linear transcripts (for review see [75,76]. There is increasing evidence that circRNAs play important roles in the development of several cancers [75,77,78]; however, information regarding their involvement in HPV-driven cancer and their potential prognostic role still remains significantly limited to CSCC (Table 2). In fact, although the role of circRNAs in HNSCC has been recently reviewed [79], little is known about circRNA expression in HPV-driven OPSCC [59], probably because HPV status was not fully reported in all studies.

### CSCC

By using RNA-seq, Wang et al. [100] explored the expression profiles of several ncRNAs in HPV16-induced CSCC and matched adjacent non-tumor tissues from three patients. Authors identified 99 circRNAs that were differentially expressed in CSCC patients, and 44 circRNAs have not been reported before. In a subsequent study, circRNA microarray demonstrated a significant increase of circRNA8924 expression in CSCC [80]. CircRNA8924 was found to adsorb miR-518-d-5p/miR-519-5p and to promote the expression of the polycomb protein chromobox 8, which has been shown to be a key regulator of several cancers, including CSCC. In fact, circRNA8924 knockdown significantly inhibited the proliferation, migration, and invasion of HPV-positive HeLa and SiHA cell lines both in vitro and in vivo [80]. Similarly, knockdown of circ_0005576 in the HPV-positive HeLa and SiHA cells significantly reduced CSCC aggressiveness. Mechanistically, circ_0005576 facilitated CSCC progression by binding miR-153 and thereby upregulating the kinesin family member 20A [81]. In the last years, other circRNAs have been identified to participate in CSCC tumorigenesis. However, in these studies the HPV-status of CSCC tissues was not defined and/or circRNAs appeared to be deregulated also in HPV-negative CSCC cell lines, indicating their expression might not be limited to HPV infection [101,102,103,104,105].

## 5. PiRNAs and PIWI-Like Proteins Expression in HPV-Driven Cancers

PiRNAs are very similar in size to miRNAs since they are 26–30 nucleotides in length, but far exceed the total number of miRNAs; in fact, about 23,439 piRNAs have been discovered so far [76]. PiRNAs specifically associate with the PIWI proteins, a subfamily of Argonaute proteins, to exert their regulatory functions (for review see Rojas-Ríos et al. [106]). Unlike miRNAs, the piRNA/PIWI complex principally acts through epigenetic silencing rather than mRNA targeting [107]. PiRNAs guide PIWI proteins to the genomic region where they share complementarities, and regulate the epigenetic status of the target sequence by recruiting epigenetic factors required for DNA methylation and/or histone modifications [108]. Besides, the piRNA/PIWI complex can also regulate gene expression at the post-transcriptional level, via alternative splicing or regulating mRNA stability, or at the post-translational level through the binding of the coding protein [109]. Although piRNAs and PIWI proteins have not been extensively studied in cancer, a limited number of published data suggest their expression is altered in HPV-driven tumors, and associated with prognosis (Table 2).

### 5.1. CSCC

Due to restricted expression during embryonic development and in several tumor types, PIWI proteins have been suggested to act as oncogenes and/or to represent a marker of cancer stem cells [110]. Interestingly, the expression of both PIWI-like protein 1 (PIWIL1) and PIWI-like protein 2 (PIWIL2) has been observed in tissues from patients with HPV16-positive CSCC [111,112]. High PIWIL1 expression was significantly associated with CSCC invasion [111], thus supporting the interaction between HPV16 and host cells during CSCC carcinogenesis. According to these data, both in vitro and in vivo studies demonstrated that PIWIL1 expression increased tumorigenesis, resistance to chemotherapeutic drugs, and self-renewal abilities of the HPV-positive HeLa and SiHa cell lines [84]. Feng et al. reported high levels of PIWIL2 in high-grade cervical intraepithelial neoplasia (CIN) and in CSCC, whereas in healthy tissue and low-grade CIN PIWIL2 was weakly expressed [85]. Additionally, the HPV-positive HeLa, SiHA and CaSki cell lines constitutively expressed PIWIL2; in contrast, PIWIL2 expression was undetectable in the HPV-negative C33A cell line [85], indicating that PIWIL2 activation in CSCC might depend on the integration of HR-HPV DNA into the host cell genome. Consistent with this hypothesis, authors demonstrated that PIWIL2 expression was restored in human keratinocyte cells following transfection with lentivirus containing complete HPV16 E6 and E7 sequences. Furthermore, PIWIL2 overexpression significantly induced histone H3 lysine 9 (H3K9) acetylation and decreased H3K9 trimethylation, thus reprogramming human keratinocyte cells into tumor-initiating cells. On the other hand, PIWIL2 knockdown led to an upregulation of p53 and p21, and reduced the tumorigenic potential of the HPV-positive HeLa and SiHa cells both in vitro and in vivo [85]. With regard to other members of the PIWI protein family, the role of PIWI-like protein 4 (PIWIL4) has also been investigated in CSCC, and results showed that PIWIL4 expression promoted a significant increase in cell growth and proliferation, and prevented apoptosis, by inhibiting p14ARF/p53 pathway in HPV-positive HeLa cell line [86].

### 5.2. HNSCC

Aberrant expression of piRNAs has been recently observed in HNSCC samples when compared to normal tissue. In particular, Firmino et al. identified 41 piRNAs that were differently expressed between HPV-positive and -negative HNSCC. Interestingly, 11 piRNAs were deregulated in tumors positive for HPV16 or HPV18 infection and, among them, the expression of piRNAs FR018916, FR140858, FR197104, FR237180, and FR298757 was associated with worse overall survival (OS), thus highlighting their potential clinical utility in HPV-positive HNSCC [82]. Subsequently, Krishnan et al. used RNA-sequencing datasets from The Cancer Genome Atlas (TCGA) to identify 30 piRNAs that were deregulated in HPV-driven HNSCC. Among them, six piRNAs (piR-36742, piR-33519, piR-36743, piR-34291, piR-36340, piR-62011) were aberrantly expressed in smoking versus never smoking HPV-positive HNSCC patients [83], suggesting that some piRNAs may be commonly implicated in smoking-related and HPV-driven HNSCC. Similarly, PIWIL4 was aberrantly expressed in smoking with respect to never smoking HPV-positive HNSCC patients as well [83]. Interestingly, piR-36743 was previously identified to be implicated in breast cancer [113], indicating the ability of the same piRNA to modulate other malignancies. Starting from the 30 HPV-deregulated piRNAs, authors further verified that the expression level of piR-30652, piR-33686, piR-36340, and piR-45029 was significantly associated with higher histologic grade, with piR-30652 being significantly predictive of patient outcome in both univariate and multivariate regression analyses.

### 5.3. SCCA

To date, no information on piRNAs/PIWIs involvement in HPV-driven SCCA is available, and the existing data concerning other sncRNAs (i.e., miRNAs) is also extremely scarce [114,115]. Therefore, there are many unknown questions about piRNAs/PIWIs that need to be explored in HPV-related tumors, especially in HPV-positive SCCA.

## 6. LncRNAs Expression in HPV-Driven Cancers

LncRNAs were initially identified as mRNA-like transcripts that do not code for proteins since they are in many ways very similar to mRNAs, including their biogenesis. However, a further characterization of lncRNAs has allowed to distinguish them from other major classes of RNA transcripts (for review see Karapetyan et al. [116]). LncRNAs can be subdivided according to their biogenesis loci in sense, antisense, bidirectional, intergenic, and intronic lncRNAs (for review see Rinn et al. [117]). Sense lncRNAs are transcribed in the same direction of exons, and they may overlap with introns and part or the entire sequence of protein-coding genes [118]. Besides representing functional RNA molecules able to regulate gene expression, sense lncRNAs can translate into protein [118]. Antisense lncRNAs are transcribed from the antisense strand of protein-coding genes, whereas bidirectional lncRNAs are expressed from the promoter of a protein-coding gene, but in the opposite direction [118]. Intergenic lncRNAs (lincRNAs) originate from the region between two protein-coding genes, and have been found to associate with chromatin modifying proteins [119]. Finally, intronic lncRNAs can be transcribed from an intronic region of a protein-coding gene in the sense or antisense direction [118]. LncRNAs play important roles in various cellular processes since their functions are highly pleiotropic; in fact, lncRNAs can regulate gene expression at many levels, such as epigenetic, transcriptional, post-transcriptional, translational, and post-translational [120]. Therefore, it is not surprising that upon viral infections most modifications occur in lncRNAs expression [121]. Consistently, an increasing number of studies have revealed a large amount of lncRNAs whose expression is deregulated in HPV-driven cancers, with most of them mainly focused on CSCC (Table 2). LncRNAs exhibit a more cell type-specific restricted expression pattern than protein-coding genes [122,123,124]. In addition, lncRNAs are stable in a broad range of specimen types (FFPE, plasma, and other body fluids), and are easily accessible for analysis using non-invasive methods [125,126], thus resulting appealing as prognostic/predictive biomarkers. So far, a small number of studies has highlighted the significant implication of lncRNAs as prognostic biomarkers in HPV-driven cancers (Table 2).

### 6.1. CSCC

A microarray analysis revealed that thousands of host lncRNAs had differential expression in oncogenic HPV-positive cells compared to the HPV-negative C33A cell line. In particular, 4750 lncRNAs were differentially expressed in the HPV16 positive SiHa cells compared with C33A cell line, including 2127 upregulated and 2623 downregulated lncRNAs. Similarly, 5026 lncRNAs were differentially expressed in the HPV18 positive HeLa cells respect to C33A cell line and, among these deregulated lncRNAs, 2218 were upregulated whereas 2808 were downregulated. In this study, the authors further demonstrated that HPV could exert effects on the development and progression of CSCC via altering the expression of lncRNAs and their downstream mRNAs targets [89]. In fact, in the HPV-positive SiHa cell line, the lncRNAs ENST00000503812 was upregulated whereas the expression of its target genes RAD51 paralog B (RAD51B), which is a component of the DNA double-strand break repair pathway [88], and interleukin-28A, which plays a role in immune defense against viruses [87], was decreased [89]. Interestingly, HPV integration was previously shown to disrupt RAD51B expression in CSCC [127]. Therefore, ENST00000503812 upregulation may impair DNA repair pathway and immune responses in HPV16 positive CSCC cells. In addition, ENST00000420168, ENST00000564977 upregulation and TCONS_00010232 downregulation showed a significant correlation with the increased expression of the oncogene forkhead box Q1 and the reduced expression of the apoptosis-related gene caspase-3 in the HPV-positive HeLa cell line [89]. These results indicate that HPV18 might alter ENST00000420168, ENST00000564977, and TCONS_00010232 expression in order to promote cell proliferation and to prevent apoptosis during CSCC progression. As shown by Zhou et al. [90], the lncRNA oncogene-induced senescence 1 (OIS1) was significantly downregulated in the majority of tumor tissues from HPV-positive CSCC compared with adjacent non-tumor tissues, but not in HPV-negative CSCC patients. Serum levels of OIS1 were also significantly lower in HPV-positive CSCC [90], indicating that OIS1 downregulation was specifically involved in the pathogenesis of HPV-driven CSCC. Accordingly, OIS1 overexpression markedly reduced the proliferation of the HPV-positive SiHa cells, potentially by inhibiting the expression of the mitogen-activated protein kinase kinase kinase 4 (MTK-1) [90]. MTK-1 expression was also reduced following GATA binding protein 6 antisense (GATA6-AS) overexpression in CSCC cell lines; however, GATA6-AS expression levels were revealed to be significantly reduced in both HPV-positive and -negative CSCC patients, suggesting GATA6-AS might play a role in CSCC through an HPV-independent pathway [128]. The study of Wang et al., that was already discussed above, reported 19 lncRNAs that were differentially expressed between HPV-positive CSCC and adjacent non-tumor tissues, and the co-expression network and function prediction suggested that all of them could play a role in HPV-driven CSCC [100]. The majority of these lncRNAs were intergenic, and three lncRNAs have not been described before. Among the differentially expressed lncRNAs, the authors identified urothelial cancer associated 1 (UCA1) which has gained much attention in recent years due to its aberrant expression in several cancers [100]. UCA1 has also been shown to promote cisplatin resistance, suggesting its potential use as a target for a novel therapeutic strategy in CSCC [91]. Unfortunately, the regulatory mechanism between UCA1 expression and cisplatin resistance is still unknown. Small nucleolar RNA host gene 8 (SNHG8) was recently clarified as a critical driving force for the development of HPV-positive CSCC [92]. Enhanced SNHG8 expression was found in HPV-positive CSCC cell lines but not in the HPV-negative C33A cells, thus indicating that HPV infection led to SNHG8 deregulation. In addition, SNHG8 silencing in HPV-positive HeLa and SiHa cells reduced cell proliferation and migration, and promoted apoptosis. Functional studies revealed that SNHG8 could bind to the enhancer of zeste homolog 2, thus inhibiting the transcription of the tumor suppressor reversion inducing cysteine-rich protein with kazal motifs in CSCC cells [92]. Human ovarian cancer-specific transcript 2 has also been reported to be upregulated in HPV-positive CSCC tissues and cell lines, and to act as sponge for the miRNA let-7b, thus promoting CSCC cell proliferation, migration, and invasion along with reduced apoptosis [93]. Differently, maternally expressed gene 3 (MEG3) was shown to function as a tumor suppressor in CSCC; in fact, MEG3 expression inhibited cell proliferation and promoted apoptosis in CSCC cells through modulating the level of miR-21-5p [96]. In addition, MEG3 expression was negatively correlated not only with CSCC grade and survival but also with HR-HPV infection [96]. Surprisingly, low MEG3 expression was associated with favorable prognosis in HPV-negative HNSCC [129]. However, the clinical significance of this finding is still unclear.

### 6.2. HNSCC

Using TCGA RNA-seq data from 426 HNSCC and 42 adjacent normal tissues, Nohata et al. [97] found 140 lncRNA transcripts that were significantly differentially expressed between HPV-positive and -negative tumors. Several lncRNAs were also deregulated in a panel of HPV-positive cell lines [97], and some of them have been already characterized, such as LINC01089 [130] and PTOV1-AS1 [131]. Interestingly, 19 lncRNAs were specifically expressed in HPV-positive and p53 wild type HNSCC, suggesting they might represent potential key molecules in HPV-driven oncogenesis [97]. A similar study identified eight lncRNAs that were associated with better prognoses in HPV-driven HNSCC, including lnc-IL17RA-11 whose expression promoted HNSCC cell sensitivity to radiotherapy [98]. The regulatory mechanism of lnc-IL17RA-11 upregulation has also been illustrated. HPV infection could stimulate estrogen receptor α to increase lnc-IL17RA-11 expression in order to upregulate the activity of genes involved in processes that enhance sensitivity to radiation therapy [98]. These findings might explain why HPV-positive HNSCC are more sensitive to radiotherapy. By analyzing lncRNAs profiling data and the corresponding clinic-pathologic variables of 371 HNSCC patients from TANRIC and cBioPortal, Cui et al. [132] defined a signature of 15 lncRNAs with prognostic significance for recurrence-free survival. Importantly, when HNSCC patients were stratified according to their HPV status, the 15 lncRNAs signature remained a clinically and statistically significant prognostic model. Similarly, by referring to the TCGA data available from the cBioportal and UALCAN databases, Kolenda et al. [99] provided experimental support on the association of the eosinophil granule ontogeny transcript (EGOT) lncRNA upregulation with the progression of HPV-positive HNSCC, but the exact mechanism for its involvement in HPV infection was not reported.

## 7. NcRNAs Modulated by E6/E7 Oncoproteins

Although the ability of HPV E6 and E7 oncoproteins to modulate the expression of many protein-coding or miRNA-coding genes have been well documented, their role in the regulation of ncRNA in HPV-driven cancers is still largely obscure. So far, the majority of studies have focused specifically on CSCC where both E6 and E7 HPV oncoproteins were shown to modulate the expression of several lncRNA (Figure 2). Furthermore, some of the HPV E6 and/or E7 deregulated lncRNAs have been suggested as potential prognostic biomarkers (Table 3). For instance, E6 was recently proposed to increase the expression of the cervical carcinoma expressed PCNA regulatory (CCEPR) lncRNA in CSCC [133]. A study of Yang et al. reported that CCEPR regulated CSCC cell proliferation by binding and stabilizing PCNA mRNA [134]. Accordingly, high levels of CCEPR indicated poor prognosis in HPV-positive CSCC patients [135]. However, in the study of Sharma et al. perturbation of CCEPR expression did not alter PCNA mRNA levels in CSCC cell lines [133], indicating that PCNA mRNA stabilization might not be the primary mechanism by which CCEPR modulates CSCC proliferation. Besides CCEPR, FAM83H-AS1 was established to be upregulated by HPV16 E6 oncoprotein both in primary keratinocytes and in CSCC tumor samples with the expression being involved in cellular proliferation and migration, and associated with worse OS in CSCC patients [136]. As reported by Barr et al. [136], HPV16 oncogene E6 mediated FAM83H-AS1 upregulation in a p300-dependent manner. Of note, FAM83H-AS1 expression was decreased in HPV18 positive CSCC cell lines, probably because HPV18 E6 does not have the ability to interact with p300 with high efficiency [137]. Other lncRNAs are modulated by HPV16 E6, including H19 and the growth arrest-specific transcript 5 (GAS5) [136]. H19 was found to be upregulated in CSCC tissues compared with adjacent normal tissues, and to promote sirtuin 1 overexpression by sponging miR-138-5p in HPV-positive CSCC HeLa and SiHa cell lines [138]. GAS5 has showed tumor suppressor activity in several tumors, including CSCC where its decreased expression was associated with poor prognosis [139,140], and increased proliferation, invasion, and migration of CSCC cells [139]. More recently, GAS5 overexpression was demonstrated to enhance cisplatin sensitivity in HPV-positive CSCC SiHa cells by directly targeting miR-21 and regulating Akt phosphorylation [141], and to improve the radio-sensitivity of HPV-positive CSCC SiHa cells via inducing immediate early response 3 expression by sponging miR-106b both in vitro and in vivo [142]. Interestingly, GAS5 expression might depend on another lncRNA called GAS5-AS1, but the regulatory mechanism between GAS5 and GAS5-AS1 has yet to be elucidated [143].

On the other hand, the expression of a number of lncRNAs is exclusively modulated by the HPV E7 oncoprotein. Along this line, Sharma et al. [144] reported that E7 could be involved in CSCC proliferation and metastasis through regulating the expression and function of Hox transcript antisense intergenic RNA (HOTAIR), which might represent a new marker of CSCC recurrence and poor prognosis [145]. HOTAIR possesses distinct binding domains for chromatin-modifying complexes and histone demethylases [146]. Thanks to that, HOTAIR might partially regulate the expression of several genes involved in cell proliferation, migration, invasion, and EMT in CSCC [145]. Interestingly, genetic variations within HOTAIR appeared to modify the risk of CSCC. In particular, the HOTAIR rs2366152C polymorphism was frequently reported in low HOTAIR expressing HPV-positive CSCC where it allowed miR-22 to directly bind to HOTAIR [147]. The gain of the miR-22 binding site in HOTAIR was found to be concordant with miR-22 overexpression, which led to reduced E7 expression in low HOTAIR HPV-positive CSCC cells [147]. HOTAIR expression not only characterized HPV-driven CSCC, but also negatively correlated with the proportion of myeloid-derived suppressor cells in the blood samples of patients with HPV-positive HNSCC [148]; unfortunately, a causal relationship has not yet been established. In a subsequent study, loss-of-function assays allowed to identify EWSAT1-Ewing sarcoma-associated transcript 1 (LINC00277) and LINC01101 as the most upregulated lncRNAs in HPV-positive CSCC HeLa cells transfected with siRNA-HPV18 E7 [149]. Consistent with these data, HPV-positive CSCC tissues exhibited significantly reduced expression of both lncRNAs, and low expression of LINC00277 and LINC01101 could predict poor prognostic features [149].

Additionally, several lncRNAs are specifically regulated by both HPV E6 and E7 oncoproteins, including the metastasis associated lung adenocarcinoma transcript 1 (MALAT-1) which is one of the most extensively characterized lncRNA so far. By using loss of function assays Guo et al. revealed, for the first time, the tumor promoting role of MALAT-1 in the HPV-positive CSCC CaSki cells. In fact, MALAT-1 silencing impaired the migration ability of CaSki cells, increased caspase-8, caspase-3, and Bax levels, and reduced Bcl-2 and Bcl-xL expression [150]. Subsequently, other research groups studies confirmed the oncogenic functions of MALAT-1 in HPV-driven CSCC since they found that MALAT-1 positively regulated the expression of genes involved in cell cycle regulation [151], and in cell migration [152]. Therefore, MALAT-1 knockdown in HPV-positive CSCC CaSki cells led to G1 arrest [151], and reduced invasion and metastasis both in vitro and in vivo [152]. In a study of Jang et al. [151], MALAT-1 expression was detected in 6/18 cases of HPV-positive cervical normal cells and 14/22 cases of HPV-positive cervical lesions, suggesting that HPV infection might lead to MALAT-1 activation in CSCC. Consistent with this hypothesis, MALAT-1 expression was found to augment in oral keratinocytes transfected with HPV E6/E7 oncoproteins [151]. MALAT-1 was demonstrated to act as a miRNA sponge as well. For instance, MALAT-1 appeared to contribute to CSCC progression by promoting the growth factor receptor bound protein 2 overexpression through binding and sequestering its major negative regulator miR-124 [153]. Interestingly, MALAT-1 has also been implicated in the mechanism of radio-resistance in HR-HPV-driven CSCC via sponging miR-145 [154].

Besides MALAT-1, He et al. [155] has recently reported that overexpression of HPV16/18 E6 or E7 enhanced the expression of Thymopoietin pseudogene 2 (TMPOP2) lncRNA in CSCC cells, whereas depletion of both HPV16/18 oncoproteins significantly downregulated TMPOP2. Similarly, TMPOP2 was found to regulate the expression of HPV16/18 E6 and E7, thus creating a positive feedback that synergistically sustained CSCC. As shown by authors, the mechanism by which HPV16/18 E6 or E7 enhanced TMPOP2 expression was predominately governed by their capacity to promote the degradation of p53; in fact, p53 was demonstrated to bind TMPOP2 promoter and to repress its transcription. Once expressed, TMPOP2 was able to sequester HPV E6/E7-targeting miR-375 and miR-139, allowing the expression of HPV oncoproteins [155]. Besides genomic ncRNAs, human cells express a unique family of mitochondrial long noncoding RNAs (ncmtRNAs) which comprises sense (SncmtRNA) and two antisense (ASncmtRNA-1 and ASncmtRNA-2) transcripts containing long inverted repeats linked to the 5′ end of the 16S mitochondrial rRNA [156,157]. SncmtRNA and ASncmtRNAs exit the mitochondria and localize to the cytosol and to the nucleus, where they associate with chromatin and nucleoli [158]. SncmtRNA represents a marker of cell proliferation since it is expressed in normal proliferating and in tumor cells but not in resting cells [156,157]. Similarly, ASncmtRNAs are expressed in normal proliferating cells, but they are downregulated in several tumors [157], thus suggesting they might function as tumor suppressor. Immortalization of keratinocytes with the complete genome of HPV16 and HPV18 downregulated ASncmtRNAs and induced a novel sense ncmtRNAs called SncmtRNA-2. Interestingly, although ASncmtRNAs expression was shown to depend on HPV E2 oncoprotein both ASncmtRNA-1 and -2 were downregulated in the HPV-positive Hela and SiHa cell lines which did not express E2 [159]. On the other hand, SncmtRNA-2, whose expression was promoted by E6 and E7 oncoproteins, was not upregulated in Hela and SiHa cells [159], thus suggesting that other cellular factors may be involved in the regulation of ASncmtRNAs and SncmtRNA-2 after HPV transformation.

In addition to lncRNAs, it has been recently reported that HPV16 E7 oncoprotein altered the expression profiles of circRNAs in CSCC cells. In this study, HPV E7 expression altered the expression of 526 circRNAs; among them, 352 were upregulated whereas 174 were downregulated. Subsequent bioinformatic analyses indicated that differently expressed circRNAs were likely to be involved in the mTOR signaling pathway, proline metabolism, and glutathione metabolism [160].

## 8. HR-HPV-Derived NcRNAs

Besides to promote aberrant ncRNAs expression, HPV has been shown to encode its own ncRNAs. For instance, a recent study revealed that HR-HPV produced circE7, a circRNA m6A modified, preferentially localized to the cytoplasm, and associated with polysomes [59]. Zhao et al. [59] demonstrated that circE7 represented only 1%–3% of total E7 transcripts but, although its weak expression, it was critically involved in HPV-induced carcinogenesis since it was translated to produce E7 oncoprotein. Accordingly, the disruption of circE7 in HPV-positive CSCC CaSki cells reduced E7 protein levels and inhibited cancer cell growth both in vitro and in vivo. CircE7 could be detected in TCGA RNA-Seq data from HPV-positive HNSCC and CSCC [59], thus suggesting it might be used as a molecular biomarker for the presence of HR-HPV and/or as a potential prognostic indicator of clinical outcome in these patients. CircE7 expression was also found in HPV-driven SCCA [161]. Of note, HPV-positive SCCA with high levels of circE7 showed a trend towards improved survival respect to those with low or absent circE7 [161] that could be probably due to a strong E7-specific immune response.

## 9. NcRNAs as Potential Diagnostic Biomarkers in HPV-Driven Cancers

Consistent with increasing role of ncRNAs in HPV-driven cancers, a number of studies have reported their potential value as diagnostic biomarkers (Table 4). For instance, MEG3 emerged as a powerful tool for prediction of tumor size and lymph node metastasis in patients with CSCC [94]. Of note, low MEG3 expression correlated with MEG3 promoter hypermethylation in both tissues [94] and plasma [95] from CSCC patients. Starting from this evidence, Zhang et al. investigated the diagnostic power of plasma MEG3 methylation with favorable results; in fact, plasma MEG3 methylation had high power to discriminate high-grade CIN patients from healthy controls, and to predict HR-HPV infection and lymph node metastasis [95]. Serum OIS1 was also proven to be an effective diagnostic biomarker in patients with HPV-positive CSCC since it effectively distinguished them from healthy controls [90].

## 10. Therapeutic Targeting of NcRNAs

Given their stability and distinct cytoplasmatic localization, ncRNAs can be used as novel therapeutic molecular tools for the treatment of HPV-driven cancers. At present, a number of RNA-based approaches have been developed to target ncRNAs, including antisense oligonucleotides (ASOs) or siRNAs (for review see Bajan et al. [162]). In this context, treatment with MALAT-1 specific ASO decreased the size and the number of tumor nodules in a pulmonary metastasis model of human lung cancer [163]. Similarly, ASO-mediated knockdown of MALAT-1 expression resulted in slower tumor growth and metastasis reduction in a mouse mammary carcinoma model [164]. More recently, Kim et al. [165] developed nanocomplexes carrying siRNAs against MALAT-1 that efficiently enhanced sensitivity of glioblastoma tumor cells to temozolomide both in vitro and in vivo. In addition to ASO and siRNAs, circRNAs targeting HPV-related RNAs and/or RNA-binding proteins may represent another promising therapeutic approach. For instance, Jost et al. [166] produced an artificial circRNA that efficiently sequestered miRNA-122 in in vitro experiments, thereby inhibiting the propagation of Hepatitis C Virus. These results suggest that RNA-based strategies may improve prognosis and therapeutic response in patients affected by HPV-driven tumors. However, since many ncRNAs are located in the nucleus [167], it should be difficult to achieve their knockdown by using RNA-based approaches. In this case, the clustered regularly interspaced short palindromic repeats (CRISPR)-associated nuclease 9 (CRISPR/Cas9) technology would provide the best option to achieve ncRNA-related genome editing since it directly targets the genomic DNA (for review see Yang et al. [168]). Given these considerations, it is expected that future studies focused on the CRISPR/Cas9 system for editing ncRNAs will receive increased interest.

Although high-throughput technologies have recently enabled the development of small molecular compounds that may potentially inhibit ncRNAs in the coming future [169,170], several studies have demonstrated that existing drugs may also modulate ncRNAs expression. Along this line, Xia et al. revealed that metformin treatment decreased tumor growth and angiogenesis of HPV-positive CSCC cell lines, that was likely to depend on the reduced binding of MALAT-1 to the tumor suppressor miR-142-3p [171]. Besides metformin, a novel chemotherapeutic compound namely Casiopeina II-gly (Cas-II-gly) modulated MALAT-1 expression in HPV-positive CSCC cell lines. By acting on MALAT-1, Cas-II-gly inactivated Wnt pathway, thus inhibiting cell proliferation and promoting apoptosis in HPV-positive HeLa and CaSki CSCC cell lines [172]. Conversely, demethylation of MEG3 promoter by using 5-aza-2-deoxycytidine upregulated MEG3 expression and reduced proliferation of HPV-positive HeLa and CaSki cells, indicating the potential use of epigenetic drugs in HPV-driven cancers [94]. At present, other therapeutic agents have been exploited against MEG3, GAS5, and HOTAIR [173,174,175,176], but their effect in HPV-driven cancers still remain to be defined.

## 11. Conclusions

In addition to miRNAs, a huge list of ncRNAs has been identified in HPV-driven cancers so far. In particular, the use of high-throughput sequencing technologies, along with loss- and gain-of-function assays, has demonstrated that the expression of circRNAs, piRNAs, and lncRNAs promoted tumorigenesis and progression of HPV-positive cancers, thus suggesting they may be partially responsible for the clinical behavior of these tumors. Despite these findings, the functional relevance of these ncRNAs in HPV-driven cancers remains rather incomplete, in particular in SCCA where the field of ncRNAs is still at its infancy. Based on these considerations, more efforts will be necessary to profile ncRNAs in each HPV-driven cancer type. Furthermore, it will be crucial to better define molecular mechanisms underlying the association between aberrant ncRNAs expression and HPV infection, and to fully explore ncRNAs that are directly generated from HPV. In this context, in vivo experiments that more closely recapitulate the tumor microenvironment will be fundamental.

Numerous studies have documented that ncRNAs expression is tissue and cancer-specific, suggesting that ncRNAs that are linked to HPV infection could be useful in the early detection of HPV-driven cancers. More importantly, ncRNAs that show aberrant expression in both HPV-positive cancer tissues and biological fluids (i.e., plasma and/or saliva) may have a clinical utility in the non-invasive liquid biopsy approach for monitoring cancer progression and its treatment response. Despite these findings, the applicability of ncRNAs as diagnostic and prognostic biomarkers will require additional studies with larger sample sizes.

HPV-related ncRNAs have been found to be involved in tumor resistance to chemotherapy and/or radiotherapy, thus indicating they may also provide an important step towards personalized treatment, in particular for HPV-driven cancers at high risk of recurrence. As mentioned above, ncRNAs can be directly targeted by RNA-based approaches, but reliable methods for their delivery to tumor cells are needed. CRISPR/Cas9-genome editing or small drug inhibitors will also offer an exceptional opportunity to explore ncRNAs as druggable molecules. However, off-target effects and toxicities should be carefully evaluated before their clinical application. Therefore, although ncRNAs seem to be therapeutically promising, additional in vitro and in vivo preclinical studies are mandatory to design novel and more effective targeted therapies for the treatment of HPV-driven cancers.

## Figures and Tables

**Figure 1 cancers-12-01246-f001:**
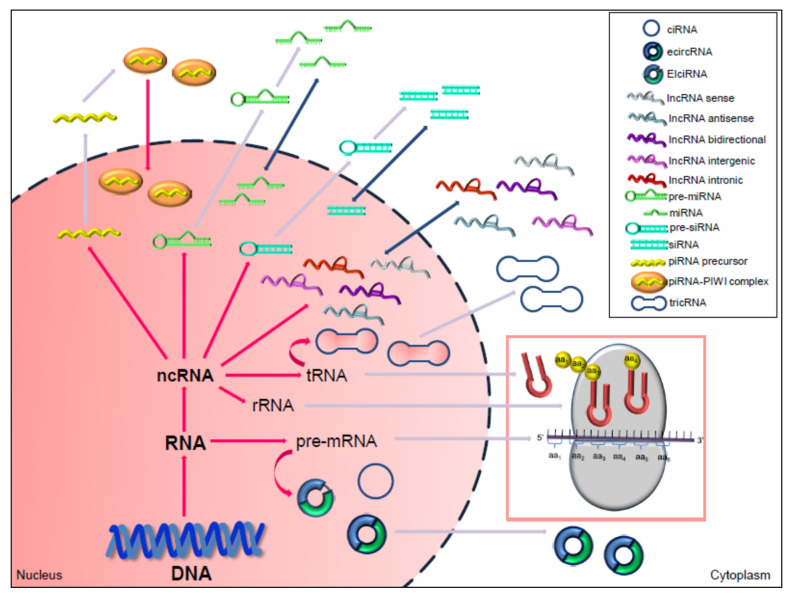
Coding and non-coding classes of RNA. Precursor messenger RNA (pre-mRNA) gives rise to mRNA, which is further translated into protein. Non-coding RNAs (ncRNAs) comprise transfer RNA (tRNA), ribosomal RNA (rRNA), and a large variety of regulatory ncRNAs, including micro RNAs (miRNAs), P-element-induced wimpy testis (PIWI)-interacting RNAs (piRNAs), small interfering RNAs (siRNAs), long non-coding RNAs (lncRNAs) and circular RNAs (circRNAs). Mature miRNAs and siRNAs are transcribed as precursors that undergo a series of nuclear and cytoplasmic processing events, and act in both nucleus and cytoplasm. Similarly, piRNAs are generated from long single-stranded piRNA precursors that are exported in the cytoplasm where they are processed; mature piRNAs are then transported into the nucleus in complex with PIWI. Most circRNAs that derive from linear pre-mRNAs, and are classified in exon-derived circRNAs (ecircRNAs), containing only exons and completely lacking introns, circular intronic RNAs (ciRNAs), which consists of only introns, exon-intron cirRNAs (EIciRNAs), in which one intron is inserted between two exons. RNA circularization can also occur through tRNA intron splicing thus generating tRNA intronic circRNAs (tricRNAs). CiRNAs and EIciRNAs are mainly nuclear, whereas ecircRNAs and tricRNAs are synthesized in the nuclear compartment and then exported to the cytosol. The lncRNAs biogenesis is mostly similar to mRNA, but they are located in the nucleus or cytoplasm, and rarely encode proteins. LncRNAs are classified as sense, antisense, bidirectional, intronic, or intergenic with respect to nearby protein-coding genes.

**Figure 2 cancers-12-01246-f002:**
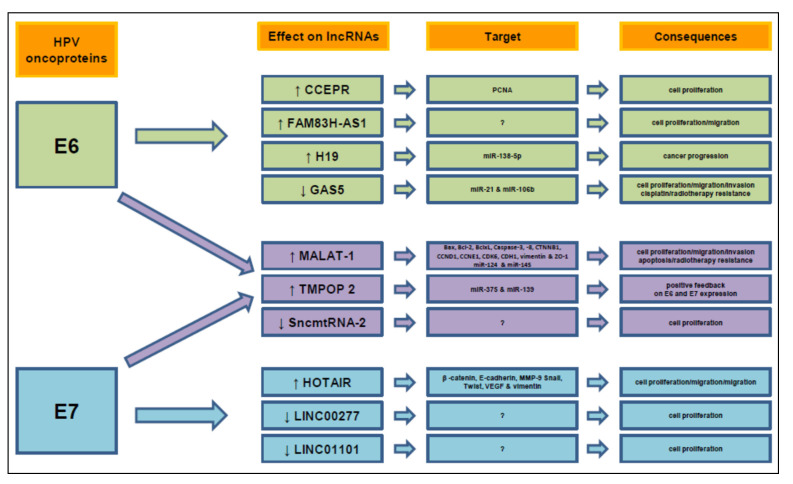
Schematic diagram of Human Papilloma Virus (HPV) E6/E7 oncoproteins affecting long non-coding RNAs (lncRNAs) expression in cervical squamous cell carcinoma. Bax, bcl-2-like protein 4; BclxL, B-cell lymphoma-extra large; Bcl-2, B-cell lymphoma 2; CCEPR, cervical carcinoma expressed PCNA regulatory; CDH1, E-cadherin; CDK6, cell division protein kinase 6; CCND1, cyclinD1; CCNE1, cyclinE; CTNNB1, b-catenin; FAM83H-AS1, FAM83H Antisense RNA 1; GAS5, growth arrest-specific transcript 5; HOTAIR, HOX transcript antisense RNA; MALAT-1, metastasis associated lung adenocarcinoma transcript 1; MMP-9, metalloproteinase-9; PCNA, proliferating cell nuclear antigen; TMPOP2, Thymopoietin pseudogene 2; VEGF, vascular endothelial growth factor; ZO-1, zonula occludens-1.

**Table 1 cancers-12-01246-t001:** Worldwide burden of cancer attributable to Human Papilloma Virus (HPV) by site.

Tumor Site	Predominant HPV Types ^*^	HPV Attributable Fraction (%)	New Cases Attributable to HPV	Prognostic Significance of HPV-Positivity	References
Head and neck cancer					
Oropharynx	HPV16; HPV33; HPV35	30.1	42,000	Better survival	[1,6,7,8]
Oral cavity	HPV16; HPV52; HPV35	2.2	5900	Inconclusive	[1,6,7,8]
Larynx	HPV16; HPV31; HPV33	2.4	4100	Inconclusive	[1,6,7,8]
Cervical cancer	HPV16; HPV18; HPV45	100	570,000	-	[1,6,9]
Anal cancer	HPV16; HPV18	88.0	29,000	Better prognosis in men	[1,6,10]
Penile cancer	HPV16; HPV6; HPV18	50.0	18,000	Inconclusive	[1,6,11,12]
Vulval cancer	HPV16; HPV33	24.9	11,000	Better survival	[1,6,13,14]
Vaginal cancer	HPV16; HPV18; HPV73	78.0	14,000	-	[1,6,15]

* HPV16 is by far the most predominant type in all HPV-driven cancer

**Table 2 cancers-12-01246-t002:** Non-coding RNA and PIWI-like proteins expression in Human Papilloma Virus (HPV)-driven cancers.

**HPV-Driven Cancer Type**	**CircRNAs ID**	**Sample Source**	**Expression Change**	**Function/Effect**	**Targets**	**Prognostic Value**	**Notes**	**References**
CSCC	CircRNA8924	Tissues, Cell lines	Up	Promote proliferation, cell cycle progression, migration, and invasion	MiR-518-d-5p/miR-519-5p			[80]
	Circ_0005576	Tissues, Cell lines	Up	Promote tumor progression	MiR-153	Correlated with advanced FIGO stage, lymph node metastasis and poor prognosis		[81]
**HPV-Driven Cancer Type**	**PiRNAs ID**	**Samples Source**	**Expression Change**	**Function/Effect**	**Targets**	**Prognostic Value**	**Notes**	**References**
HNSCC	FR018916, FR140858, FR197104, FR237180, FR298757	TCGA	Down			PiRNA expression signature can predict OS in HPV positive patients	Downregulated in HPV16/18 HNSCC samples compared to cases harboring other HPV types	[82]
	PiR-36742	TCGA	Up				Deregulated in smoking HPV-positive patients	[83]
	PiR-33519		Up					
	PiR-36743		Up					
	PiR-34291		Up					
	PiR-36340		Down					
	PiR-62011		Down					
	PiR-30652	TCGA	Up			Significantly predictive of patient outcome	Significantly associated with higher histologic grade	
	PiR-33686		Up					
	PiR-36340		Down					
	PiR-45029		Down					
**HPV-Driven Cancer Type**	**PIWI-Like Proteins ID**	**Samples Source**	**Expression Change**	**Function/Effect**	**Targets**	**Prognostic Value**	**Notes**	**References**
CSCC	PIWIL1	Tissues, Cell lines	Up	Promote tumorigenesis and tumor progression, suppress chemotherapy sensitivity				[84]
	PIWIL2	Tissues, Cell lines	Up	Promote tumorigenesis, induce H3K9 acetylation and reduce H3K9 trimethylation			PIWIL2 activation in CSCC appears to depend on the integration of HR-HPV DNA	[85]
	PIWIL4	Tissues, Cell lines	Up	Promote proliferation, inhibit apoptosis	P14ARF/p53 pathway			[86]
HNSCC	PIWIL4	TCGA consortium	Up				Upregulated in smoking HPV-positive patients	[83]
**HPV-Driven Cancer Type**	**LncRNAs ID**	**Samples Source**	**Expression Change**	**Function/Effect**	**Targets**	**Prognostic Value**	**Notes**	**References**
CSCC	ENST00000503812	Cell lines	Up	Impair DNA repair, induce immune response			Negatively correlated with RAD51B and IL-28A expression	[87,88]
	ENST00000420168, ENST00000564977	Cell lines	Up	Promote proliferation, inhibit apoptosis			Correlation with the increased expression of FOX Q1 and the reduced expression of caspase-3	[89]
	TCONS_00010232	Cell lines	Down					
	OIS1	Tissues, Serum Cell lines	Down	Suppress cell proliferation	MTK-1	Significant association between OIS1 serum levels and tumor size		[90]
	UCA1	Cell lines	Up	Induce cisplatin resistance and inhibit apoptosis				[91]
	SNHG8	Cell lines	Up	Promote proliferation and migration, inhibit apoptosis	EZH2			[92]
	HOST2	Tissues, Cell lines	Up	Promote proliferation, migration and invasion, inhibit apoptosis	MiRNA let-7b			[93]
	MEG3	Tissues, Serum, Cell lines	Down	Suppress proliferation, promote apoptosis	MiR-21-5p	Correlated with increased tumor size, advanced FIGO stage, lymph node metastasis, HPV infection, recurrence-free survival and OS		[94,95,96]
HNSCC	LINC01089	TCGA	Up					[97]
	PTOV1-AS1	TCGA	Up					[97]
	IL17RA-11	TCGA Cell lines	Up	Induce radiotherapy sensitivity		Correlated with better prognosis	HPV infection stimulates ERα to increase lnc-IL17RA-11 expression; this finding suggests why HPV-positive HNSCC are more sensitive to radiotherapy	[98]
	EGOT	TCGA data	Up	Promote tumor progression			EGOT expression levels vary according to age, N-stage, grade, location, lymph node dissection, and HPV16 status	[99]

CircRNAs, circular RNAs; CSCC, cervical squamous cell carcinoma; EGOT, eosinophil granule ontogeny transcript; ERα, estrogen receptor α; EZH2, enhancer of zeste homolog 2; FIGO, International Federation of Gynecology and Obstetrics; FOXQ1, oncogene forkhead box Q1; HNSCC, head and neck squamous cell carcinoma; HOST2, human ovarian cancer-specific transcript 2; HR-HPV, high risk HPV; H3K9, Histone H3 Lysine 9; IL17RA-11, Interleukin 17 receptor A; IL-28A, interleukin 28A; LncRNAs, long non-coding RNAs; MEG3, maternally expressed gene 3; MTK-1, mitogen-activated protein kinase kinase kinase 4; OIS1, oncogene-induced senescence 1; OS, overall survival; PiRNAs, PIWI-interacting RNAs; PIWIL, P-element-induced wimpy testis-like protein; PTOV1-AS1, PTOV1 antisense RNA 1; RAD51B, RAD51 paralog B; SNHG8, small nucleolar RNA host gene 8; TCGA, The Cancer Genome Atlas; UCA1, Urothelial Cancer Associated 1.

**Table 3 cancers-12-01246-t003:** Prognostic value of long non-coding RNAs (lncRNAs) modulated by Human Papilloma Virus E6/E7 oncoproteins in cervical squamous cell carcinoma.

LncRNAs ID	Sample Description	Expression Change	Prognostic Value	References
CCEPR	Cell lines, Tissues	Up	Positively correlated with advanced FIGO stage, lymph node metastasis, HPV infection, and poor prognosis	[133,134,135]
FAM83H-AS1	Cell lines, Tissues	Up	Poor prognosis	[136]
GAS5	Cell lines, Tissues	Down	Poor prognosis	[139,140]
HOTAIR	Cell lines, Tissues	Up	Disease recurrence and poor prognosis	[145]
	Cell lines, Tissues		Rs2366152C polymorphism associates to reduced HOTAIR expression and CSCC metastatic molecular signatures	[147]
LINC00277, LINC01101	Cell lines, Tissues	Down	Poor prognosis	[149]

CCEPR, cervical carcinoma expressed PCNA regulatory; CSCC, cervical squamous cell carcinoma; FAM83H-AS1, FAM83H Antisense RNA 1; FIGO, International Federation of Gynecology and Obstetrics; GAS5, growth arrest-specific transcript 5; HOTAIR, HOX transcript antisense RNA.

**Table 4 cancers-12-01246-t004:** Diagnostic value of long non-coding RNAs (lncRNAs) in cervical squamous cell carcinoma.

LncRNAs ID	Cohort Size		Source of LncRNAs	Sensitivity	Specificity	AUC	Diagnostic Value	References
MEG3	72 cases and 72 normal tissues		Tissues	56.1%	80.6%	0.745	Tumor-size <4 cm or ≥ 4 cm	[94]
				70.5%	67.9%	0.716	Lymph node metastasis	
	108 cases			54.8%	84.8%	0.753	Tumor-size <4 cm or ≥ 4 cm	
				76.1%	85.4%	0.862	Lymph node metastasis	
MEG3 methylation	160 CIN I-III, 168 cases, and 328 healthy patients divided into training set and test set randomly and averagely	Training set	Plasma	73.7%	94.7%	0.831	CIN III	[95]
				75.8%	88.9%	0.815	HR-HPV infection	
				93.3%	51.9%	0.741	Lymph node metastasis	
		Test set		84.2%	52.6%	0.788	CIN III	
				78.1%	70.0%	0.730	HR-HPV infection	
				82.4%	72.0%	0.804	Lymph node metastasis	
OIS1	22 HPV-negative patients, 70 HPV-positive patients, and 40 healthy patients		Plasma	n.a. *	n.a. *	0.9207	Serum OIS1 may be used to diagnose HPV-positive, but not HPV-negative CSCC	[90]

AUC, area under the curve; CIN, cervical intraepithelial neoplasia; HR-HPV, high risk Human Papilloma Virus; MEG3, maternally expressed gene 3; OIS1, oncogene-induced senescence 1. * Data are not available.

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
