# Peer review of "Beyond MicroRNAs: Emerging Role of Other Non-Coding RNAs in HPV-Driven Cancers"

_cancers, 2020, doi:10.3390/cancers12051246_

Round 1
Reviewer 1 Report
This is a very comprehensive review on ncRNAs involvement in HPV driven cancers.
Comments:
Table 1 is not necessary doesn't add anything to the review a summary table of HPV in different cancers would have been more useful
Figure 1 is not jut useless but it is wrong. Based on the figure pre-mRNAs give rise to tRNAs and rRNAs. This is not correct. Also, besides cRNAs not any other ncRNA species maturation have been shown but the review discuss many of those RNAs in the cancer context.
I generally suggest to rethink the visual presentation and add some more informative figures that are for instancing show ncRNA interactions with HPV RNAs/proteins. Visualizing some proven intaractions etc.
Minor:
6.1 should be written with italics
Author Response
May 4, 2020
Ref: Revised version of Manuscript # Cancers-771105 Version 1
Dear Reviewers,
Thank you very much for the review of our manuscript entitled: "Beyond microRNAs: emerging role of other non-coding RNAs in HPV-driven cancers" by M Casarotto et al. We highly appreciate your constructive and valuable comments and suggestions.
Following is a point-by-point reply to your comments, together with the changes that have been made in the new version of the manuscript.
Reviewer # 1
- Table 1 is not necessary doesn't add anything to the review; a summary table of HPV in different cancers would have been more useful.
Our comment: As suggested by the Reviewer, a summary table of HPV in different cancers has been provided.
- Figure 1 is not jut useless but it is wrong. Based on the figure pre-mRNAs give rise to tRNAs and rRNAs. This is not correct. Also, besides cRNAs not any other ncRNA species maturation have been shown but the review discuss many of those RNAs in the cancer context. I generally suggest to rethink the visual presentation and add some more informative figures that are for instancing show ncRNA interactions with HPV RNAs/proteins. Visualizing some proven intaractions etc.
Our comment: Thanks to the Reviewer’s observation, we realized there were some mistakes in Figure 1. Therefore, Figure 1 has been corrected and edited in order to schematically illustrate maturation of ncRNAs that are discussed in the text (miRNAs, siRNAs, piRNAs, circRNAs and lncRNAs). NcRNAs cellular localization has also been indicated. In addition, a new figure, Figure 2, schematizing the HPV oncoproteins that affect lncRNAs expression in CSCC has been added to the manuscript.
We sincerely hope that you will consider this revised version of the manuscript suitable for publication in Cancers, and I look forward to hearing from you at your earliest convenience.
Kind regards,
Elisabetta Fratta, PhD
Corresponding author
Reviewer 2 Report
Comments to the Author(s):
The review by Casarotto et al. shows the importance of different types of non-coding RNAs such as long non-coding RNAs, circular RNAs, and PIWI-interacting RNAs in the process of carcinogenesis in HPV-related cancers. After a significant literature search, the authors tried to convince the readers that recently discovered non-coding RNAs are key molecular components involved in the tumorigenesis of HPV-related cancers such as cervical, oropharyngeal, and anal cancers. Furthermore, they suggest that future research in this field will be important for diagnostic and therapeutic purposes in HPV-related cancers. Even though there are some few missing references and corrections in some tables and figures, the authors present a valuable review that will help researchers understand the importance of non-coding RNAs in the field of cancer virology.
These are some of the issues found in this review:
- Line 62-65, “…Notably, a gender-specific trend has also emerged for HPV-driven OPSCC. In fact, mirroring the downward trend of CSCC due to HPV vaccination programs, the HPV-driven OPSCC incidence is expected to decline in women, whereas the incidence among men has been increasing over the last years.” What about immune system gender differences? (Clocchiatti A, et al. 2016. Sexual dimorphism in cancer. Nat Rev Cancer) DOI:10.1038/nrc.2016.30.
- Figure 1, authors need to modify the figure to clearly show that some of these non-coding RNAs are cytoplasmic, nuclear or both. For example, snRNA and snoRNAs are exclusively nuclear, lncRNAs can be nuclear or cytoplasmic, exonic circular RNAs are mostly cytoplasmic meanwhile intronic circular RNAs are mostly nuclear.
- Table 2 and 3, authors need to add another column in these tables to give the reader the information of what type of non-coding RNA are they referring to. For example, they have some IDs like ENST00000503812, is this considered a long non-coding RNA?
- Line 343, “…By suppressing p53 activity, HPV is able to bypass cellular senescence. On the other hand, the release of E2F transcription factors allows for unscheduled cell proliferation.” Please add this reference: (Horner SM, et al. 2004. Repression of the human Papillomavirus E6 Gene Initiates p53-Dependent, Telomerase-Independent Senescence and Apoptosis in HeLa Cervical Carcinoma Cells. J Virol) DOI: 10.1128/jvi.78.8.4063-4073.2004.
- Line 363-366, “As regards the relationship between transforming HPV infection and the expression pattern of host ncRNA, numerous studies, mainly focused on the study of miRNA, have shown a different miRNA expression in HPV-positive tumor cells compared to the negative counterpart” please add these two important references (the first two manuscripts showing interactions between HPV and microRNAs): (Martinez I, et al. 2008. Human papillomavirus type 16 reduces the expression of microRNA-218 in cervical carcinoma cells. Oncogene) DOI: 10.1038/sj.onc.1210919, (Wang X, et al. 2008. Aberrant Expression of Oncogenic and Tumor-Suppressive MicroRNAs in Cervical Cancer Is Required for Cancer Cell Growth. Plos One) DOI: 10.1371/journal.pone.0002557.
- Line 491, “…human cells express a unique family of long noncoding RNAs (ncmtRNAs)…”, the authors missed to add the word mitochondrial: “…human cells express a unique family of mitochondrial long noncoding RNAs (ncmtRNAs)…”
- Line 510, typographical error. Missing the end of the parenthesis: “…siRNAs (for review see [160]).
Author Response
May 4, 2020
Ref: Revised version of Manuscript # Cancers-771105 Version 1
Dear Reviewers,
Thank you very much for the review of our manuscript entitled: "Beyond microRNAs: emerging role of other non-coding RNAs in HPV-driven cancers" by M Casarotto et al. We highly appreciate your constructive and valuable comments and suggestions.
Following is a point-by-point reply to your comments, together with the changes that have been made in the new version of the manuscript.
Reviewer # 2
- Line 62-65, “…Notably, a gender-specific trend has also emerged for HPV-driven OPSCC. In fact, mirroring the downward trend of CSCC due to HPV vaccination programs, the HPV-driven OPSCC incidence is expected to decline in women, whereas the incidence among men has been increasing over the last years.” What about immune system gender differences? (Clocchiatti A, et al. 2016. Sexual dimorphism in cancer. Nat Rev Cancer) DOI:10.1038/nrc.2016.30.
Our comment: We thank the Reviewer for highlighting this point. Introduction has been integrated with a short paragraph associating immune system gender differences with HPV-driven OPSCC incidence (lines 75-83): “One possible explanation could lie in the profound differences observed in male versus female immune responses in cancer since it has become increasingly evident that the major susceptibility of women to a variety of autoimmune diseases might contribute to enhanced immune surveillance against various tumor types [29]. Sex hormones can also affect the immune system since high estrogen levels have been shown to promote antibody production, whereas androgens have been reported to suppress immune function [30]. Consistent with these evidences, only a small proportion of seroconversions occur in men following HPV infection [31], and HPV seroprevalence in men is significantly lower than that reported among women [32]."
- Figure 1, authors need to modify the figure to clearly show that some of these non-coding RNAs are cytoplasmic, nuclear or both. For example, snRNA and snoRNAs are exclusively nuclear, lncRNAs can be nuclear or cytoplasmic, exonic circular RNAs are mostly cytoplasmic meanwhile intronic circular RNAs are mostly nuclear.
Our comment: According to the Reviewer’s comment, Figure 1 has been modified to clearly show ncRNAs nuclear and/or cytoplasmatic localization. Furthermore, Figure 1 has been edited in order to be focused exclusively on ncRNAs than are cited in the manuscript, thus excluding snRNAs and snoRNAs.
- Table 2 and 3, authors need to add another column in these tables to give the reader the information of what type of non-coding RNA are they referring to. For example, they have some IDs like ENST00000503812, is this considered a long non-coding RNA?
Our comment: Upon the Reviewer suggestion, we inserted a column that indicates the type of ncRNAs we are referring to in Table 2. Table 3 has been modified and in the revised version provides only lncRNAs information. We then confirmed that ncRNAs with IDs like ENST00000503812 are considered lncRNAs.
- Line 343, “…By suppressing p53 activity, HPV is able to bypass cellular senescence. On the other hand, the release of E2F transcription factors allows for unscheduled cell proliferation.” Please add this reference: (Horner SM, et al. 2004. Repression of the human Papillomavirus E6 Gene Initiates p53-Dependent, Telomerase-Independent Senescence and Apoptosis in HeLa Cervical Carcinoma Cells. J Virol) DOI: 10.1128/jvi.78.8.4063-4073.2004.
Our comment: As suggested by the Reviewer, the reference “Horner SM, et al. 2004. Repression of the human Papillomavirus E6 Gene Initiates p53-Dependent, Telomerase-Independent Senescence and Apoptosis in HeLa Cervical Carcinoma Cells. J Virol” has been added to the text.
- Line 363-366, “As regards the relationship between transforming HPV infection and the expression pattern of host ncRNA, numerous studies, mainly focused on the study of miRNA, have shown a different miRNA expression in HPV-positive tumor cells compared to the negative counterpart” please add these two important references (the first two manuscripts showing interactions between HPV and microRNAs): (Martinez I, et al. 2008. Human papillomavirus type 16 reduces the expression of microRNA-218 in cervical carcinoma cells. Oncogene) DOI: 10.1038/sj.onc.1210919, (Wang X, et al. 2008. Aberrant Expression of Oncogenic and Tumor-Suppressive MicroRNAs in Cervical Cancer Is Required for Cancer Cell Growth. Plos One) DOI: 10.1371/journal.pone.0002557.
Our comment: As suggested by the Reviewer, the references “Martinez I, et al. 2008. Human papillomavirus type 16 reduces the expression of microRNA-218 in cervical carcinoma cells. Oncogene”, and “Wang X, et al. 2008. Aberrant Expression of Oncogenic and Tumor-Suppressive MicroRNAs in Cervical Cancer Is Required for Cancer Cell Growth. Plos One” have been added to the text.
- Line 491, “…human cells express a unique family of long noncoding RNAs (ncmtRNAs)…”, the authors missed to add the word mitochondrial: “…human cells express a unique family of mitochondrial long noncoding RNAs (ncmtRNAs)…”
Our comment: Following Reviewer’s observation, we have added the word mitochondrial.
- Line 510, typographical error. Missing the end of the parenthesis: “…siRNAs (for review see [160]).
Our comment: Thanks to the Reviewer, we have corrected this typographical error.
We sincerely hope that you will consider this revised version of the manuscript suitable for publication in Cancers, and I look forward to hearing from you at your earliest convenience.
Kind regards,
Elisabetta Fratta, PhD
Corresponding author
Reviewer 3 Report
In the manuscript entitled “Beyond microRNAs: emerging role of other non-coding RNAs in HPV-driven cancers” by Mariateresa Casarotto et al., the authors review current knowledge about non-coding RNAs, with miRNAs exclusion, related to human papillomaviruse (HPV)-driven cancers. The manuscript collects recent results concerning ncRNA in HPV-driven cancers, it is detailed and refers to many original papers. However, there are several concerns that should be addressed.
Major concerns:
1. The manuscript lacks a guiding theme. In the introduction, the Authors emphasize a need for biomarkers and prognostic factors in HPV-driven tumors (lines: 51, 77, 89-94), but in the main part of the manuscript is hard to find out precise information which ncRNAs can be promising as a diagnostic or prognostic biomarker. The Authors should indicate which ncRNA can be included in this group and place these ncRNAs in a separate table. They should take into account their diagnostic value, sensitivity, specificity and selectivity. Deregulation of ncRNA upon HPV-driven cancerogenesis is not enough.
- The general phrases describing the relation of ncRNA and HPV-driven pathogenesis, e.g. “association”, “correlation”, “related to” are overused. Thus, it is difficult to rank ncRNA in order of their relevance and point out the crucial for HPV-driven pathogenesis, ncRNA molecules.
- What were the criteria for division of the collected material between subchapters: “circRNAs/ piRNAs/ LncRNAs expression in HPV-driven cancers”, “NcRNA and HPV-driven cancerogenesis” and “NcRNAs modulated by HR-HPV”? There is no explanation of what is the difference between ncRNAs included in Table 2 and Table 3. Can ncRNA from subchapters “circRNAs/ piRNAs/ LncRNAs expression in HPV-driven cancers” be included in “NcRNA and HPV-driven cancerogenesis” subchapter?
- In the subchapter “3.2 HNSCC” there is no single circRNA mentioned and any other detail about circRNA in HNSCC or OPSCC is provided. Some specific information should be introduced or the subchapter should be removed.
- The “4.1 CSCC” subchapter is not about “piRNA expression in HPV-driven cancers” (as it should be) but about PIWI-like proteins expression.
- In “lncRNA expression in HPV-driven cancers” subchapter the Authors recall the number of deregulated lncRNAs upon HPV-driven cancerogenesis in different studies. The numbers are disproportionate, even taking into account a distinct source of the research material. Can the authors comment on this?
- Table 2 and Table 3 – the columns “samples description” and “diagnostic potential” should be introduced. Non-coding RNAs in tables should be organized in circRNAs, lncRNAs and piRNAs clusters.
- A list that collects ncRNAs deregulated in at least two different studies should be prepared.
Minor points:
- Lines 101,102: ”NcRNAs interact with histone modifying complexes or DNA methyltransferases, being also targets of these epigenetic mediators” It is not true. NcRNAs include also e.g. miRNA, siRNAs and they have a different mechanism of action.
- Lines: 148-149. “…however, information regarding their involvement in HPV-driven cancer still remains significantly limited to CSCC”. There is some information about circRNAs in HNSCC and OSCC that can be included
- HPVs own circRNAs should be listed
- 6.2. NcRNAs modulated by E6/E7 oncoproteins – the font should be the same as in 6.1
- Information from lines 402-404: “In addition, it has been recently reported that HPV16 E7 oncoprotein altered the 402 expression profiles of circRNAs in CSCC cells. In this study, HPV E7 expression altered the 403 expression of 526 circRNAs; among them, 352 were up-regulated whereas 174 were down-regulated’, should be transferred to the next subchapter.
- In the subchapter “7. Therapeutic targeting of ncRNAs” the Authors write about “tools” and “targets”. The text should be rearranged to gather in one paragraph “tools” and in another “targets”.
Author Response
May 4, 2020
Ref: Revised version of Manuscript # Cancers-771105 Version 1
Dear Reviewers,
Thank you very much for the review of our manuscript entitled: "Beyond microRNAs: emerging role of other non-coding RNAs in HPV-driven cancers" by M Casarotto et al. We highly appreciate your constructive and valuable comments and suggestions.
Following is a point-by-point reply to your comments, together with the changes that have been made in the new version of the manuscript.
Reviewer # 3
- The manuscript lacks a guiding theme. In the introduction, the Authors emphasize a need for biomarkers and prognostic factors in HPV-driven tumors (lines: 51, 77, 89-94), but in the main part of the manuscript is hard to find out precise information which ncRNAs can be promising as a diagnostic or prognostic biomarker. The Authors should indicate which ncRNA can be included in this group and place these ncRNAs in a separate table. They should take into account their diagnostic value, sensitivity, specificity and selectivity. Deregulation of ncRNA upon HPV-driven cancerogenesis is not enough.
Our comment: Upon Reviewer suggestion we have revised the text and both Table 2 and Table 3 with the aim to better emphasize the potential prognostic role of some ncRNAs that have been reported as deregulated in HPV-driven cancers. Moreover, we added a new paragraph entitled “NcRNAs as potential biomarkers in HPV-driven cancers” in which we included the lncRNAs MEG3 and OIS1, and a table, Table 4, that reports their diagnostic value.
- The general phrases describing the relation of ncRNA and HPV-driven pathogenesis, e.g. “association”, “correlation”, “related to” are overused. Thus, it is difficult to rank ncRNA in order of their relevance and point out the crucial for HPV-driven pathogenesis.
Our comment: As suggested by the Reviewer some sentences have been rephrased as indicated below:
“indicating their expression might not be correlated to HPV infection” now is “indicating their expression might not be limited to HPV infection” (line 265);
“the expression of both PIWI-like protein 1 (PIWIL1) and PIWI-like protein 2 (PIWIL2) has been positively correlated with HPV16 infection in tissues from CSCC patients” now is “the expression of both PIWI-like protein 1 (PIWIL1) and PIWI-like protein 2 (PIWIL2) has been observed in tissues from patients with HPV16-positive CSCC” (lines 286-288);
“Interestingly, 11 piRNAs were specifically associated with HPV16 or HPV18 infection” now is “Interestingly, 11 piRNAs were specifically associated with deregulated in tumors positive for HPV16 or HPV18 infection” (lines 313-314);
“In this study, mRNAs changes were also investigated, and then correlated with lncRNAs expression [76]. In the HPV-positive SiHa cell line a significant negative correlation was found between the up-regulation of the lncRNAs ENST00000503812 and the expression of genes encoding for RAD51 paralog B (RAD51B), which is a component of the DNA double-strand break repair pathway [75], and interleukin-28A, which plays a role in immune defense against viruses [74].” now is “In this study, authors further demonstrated that HPV could exert effects on the development and progression of CSCC via altering the expression of lncRNAs and their downstream mRNAs targets [77]. In fact, in the HPV-positive SiHa cell line, the lncRNAs ENST00000503812 was up-regulated whereas the expression of its target genes RAD51 paralog B (RAD51B), which is a component of the DNA double-strand break repair pathway [76] and interleukin-28A, which plays a role in immune defense against viruses [75], was decreased [77].” (lines 373-379);
“These results indicate that HPV18 might alter lncRNAs expression” now is “These results indicate that HPV18 might alter ENST00000420168, ENST00000564977, and TCONS_00010232 expression” (lines 385-387);
“UCA1 has also been associated with cisplatin resistance” now is “UCA1 has also been showed to promote cisplatin resistance” (lines 406-407);
“indicating SNHG8 was closely related with HPV infection.” now is “thus indicating that HPV infection led to SNHG8 deregulation.” (line 412);
“A similar study identified eight lncRNAs that were correlated with better prognoses in HPV-driven HNSCC. Among them, lnc‐IL17RA‐11 exhibited the highest correlation with both HPV infection and radiotherapy response” now is “A similar study identified eight lncRNAs that were associated with better prognoses in HPV-driven HNSCC, includinglnc‐IL17RA‐11 which expression promoted HNSCC cell sensitivity to radiotherapy” (lines 435-438).
“The gain of the miR-22 binding site in HOTAIR was found to be concordant with miR-22 overexpression, but correlated negatively with E7 expression in low HOTAIR HPV-positive CSCC cells” now is “The gain of the miR-22 binding site in HOTAIR was found to be concordant with miR-22 overexpression, which led to reduced E7 expression in low HOTAIR HPV-positive CSCC cells” (lines 497-499);
“and low expression of LINC00277 and LINC01101 correlated with poor prognostic features” now is “and low expression of LINC00277 and LINC01101 could predict poor prognostic features” (lines 507-508);
“In fact, MALAT-1 silencing impaired the migration ability of CaSki cells, and associated to high levels of caspase-8, caspase-3, and Bax, along with reduced expression of Bcl-2 and Bcl-xL” now is “In fact, MALAT-1 silencing impaired the migration ability of CaSki cells, increased caspase-8, caspase-3, and Bax levels, and reduced Bcl-2 and Bcl-xL expression” (lines 513-515);
“and this effect was associated with decreased binding of MALAT-1 to the tumor suppressor miR-142-3p” now is “that was likely to depend on the reduced binding of MALAT-1 to the tumor suppressor miR-142-3p” (lines 677-678);
“the expression of circRNAs, piRNAs and lncRNAs is strictly associated with tumorigenesis and progression of HPV-positive cancer” now is “the expression of circRNAs, piRNAs and lncRNAs promoted tumorigenesis and progression of HPV-positive cancers” (lines 692-693).
- What were the criteria for division of the collected material between subchapters: “circRNAs/ piRNAs/ LncRNAs expression in HPV-driven cancers”, “NcRNA and HPV-driven cancerogenesis” and “NcRNAs modulated by HR-HPV”? There is no explanation of what is the difference between ncRNAs included in Table 2 and Table 3. Can ncRNA from subchapters “circRNAs/ piRNAs/ LncRNAs expression in HPV-driven cancers” be included in “NcRNA and HPV-driven cancerogenesis” subchapter?
Our comment: We thank the Reviewer for this input. We have modified the text and moved the “HPV-driven cancerogenesis” subchapter after the Introduction section. Furthermore, the description of HPV own circRNAs has been placed in a new section entitled: “HR-HPV-derived ncRNAs”. Table 3 has also been modified and focused on the prognostic potential of lncRNAs that have been reported to be directly modulated by E6 and/or E7 oncoproteins. Finally, a new Figure, Figure 2, has been created to schematically illustrate lncRNAs that are affected by HPV E6/E7 oncoproteins.
- In the subchapter “3.2 HNSCC” there is no single circRNA mentioned and any other detail about circRNA in HNSCC or OPSCC is provided. Some specific information should be introduced or the subchapter should be removed.
Our comment: Following Reviewer’s observation, this subchapter has been removed.
- The “4.1 CSCC” subchapter is not about “piRNA expression in HPV-driven cancers” (as it should be) but about PIWI-like proteins expression.
Our comment: According to the Reviewer’s comment, the title of chapter 4 (now chapter 5) has been edited into: “PiRNAs and PIWI-like proteins expression in HPV-driven cancers”.
- In “lncRNA expression in HPV-driven cancers” subchapter the Authors recall the number of deregulated lncRNAs upon HPV-driven cancerogenesis in different studies. The numbers are disproportionate, even taking into account a distinct source of the research material. Can the authors comment on this?
Our comment: According to the Reviewer’s observation, in the revised text, we discussed about the large number of lncRNAs that have been found deregulated in HPV-driven cancers (lines 352-358): “LncRNAs play important roles in various cellular processes since they functions are highly pleiotropic; in fact lncRNAs can regulate gene expression at many levels, such as epigenetic, transcriptional, post-transcriptional, translational, and post-translational [107]. Therefore, it is not surprising that upon viral infections most modifications occur in lncRNAs expression [108]. Consistently, an increasing number of studies have revealed a large amount of lncRNAs which expression is deregulated in HPV-driven cancers, with most of them mainly focused on CSCC (Table 2).”
- Table 2 and Table 3 – the columns “samples description” and “diagnostic potential” should be introduced. Non-coding RNAs in tables should be organized in circRNAs, lncRNAs and piRNAs clusters.
Our comment: As suggested by the Reviewer the column “samples description” has been introduced in Tables 2 and 3, whereas the diagnostic potential of ncRNAs has been included in a separate Table as discussed above (see point 1).
- A list that collects ncRNAs deregulated in at least two different studies should be prepared.
Our comment: As requested by the Reviewer, ncRNAs deregulated in at least two different studies have been listed in Supplementary Table 1.
Minor points:
- Lines 101,102: ”NcRNAs interact with histone modifying complexes or DNA methyltransferases, being also targets of these epigenetic mediators” It is not true. NcRNAs include also e.g. miRNA, siRNAs and they have a different mechanism of action.
Our comment: Following Reviewer’s observation, this sentence has been removed.
- Lines: 148-149. “…however, information regarding their involvement in HPV-driven cancer still remains significantly limited to CSCC”. There is some information about circRNAs in HNSCC and OSCC that can be included.
Our comment: According to the Reviewer’s comment, we looked for studies focused on circRNAs in HPV-driven HNSCC but we did not find anything apart from the paper of Zhao et al. that we discuss in the paragraph entitled: “HR-HPV-derived ncRNAs”. Two recent papers, “Ouyang SB et al. CircRNA_0109291 regulates cell growth and migration in oral squamous cell carcinoma and its clinical significance. Iran J Basic Med Sci. 2018 Nov;21(11):1186-1191”, and “Wang WL et al. Competing endogenous RNA analysis reveals the regulatory potency of circRNA_036186 in HNSCC. Int J Oncol. 2018 Oct; 53(4): 1529–1543”, investigated circRNAs in HNSCC, but the HPV status was not reported. Therefore, this paragraph has been modified in :” … however, information regarding their involvement in HPV-driven cancer and their potential prognostic role still remains significantly limited to CSCC (Table 2). In fact, although the role of circRNAs in HNSCC has been recently reviewed [69], little is known about circRNA expression in HPV-driven OPSCC [49], probably because HPV status was not fully reported in all studies”.
- HPVs own circRNAs should be listed
Our comment: As previously explained (see point 3), a new section entitled: “HR-HPV-derived ncRNAs” has been introduced.
- 6.2. NcRNAs modulated by E6/E7 oncoproteins – the font should be the same as in 6.1.
Our comment: This subchapter has been modified in chapter.
- Information from lines 402-404: “In addition, it has been recently reported that HPV16 E7 oncoprotein altered the 402 expression profiles of circRNAs in CSCC cells. In this study, HPV E7 expression altered the 403 expression of 526 circRNAs; among them, 352 were up-regulated whereas 174 were down-regulated’, should be transferred to the next subchapter.
Our comment: As suggested by the Reviewer, this paragraph has been moved to the subchapter entitled “NcRNAs modulated by E6/E7 oncoproteins”.
- In the subchapter “7. Therapeutic targeting of ncRNAs” the Authors write about “tools” and “targets”. The text should be rearranged to gather in one paragraph “tools” and in another “targets”.
Our comment: According to the Reviewer’s comment, the sentence has been rephrased to: “Given their stability and distinct cytoplasmatic localization, ncRNAs can be used as novel therapeutic molecular tools for the treatment of HPV-driven cancers.”
We sincerely hope that you will consider this revised version of the manuscript suitable for publication in Cancers, and I look forward to hearing from you at your earliest convenience.
Kind regards,
Elisabetta Fratta, PhD
Corresponding author
Round 2
Reviewer 3 Report
In the new version of the manuscript entitled “Beyond microRNAs: the emerging role of other non-coding RNAs in HPV-driven cancers” by Mariateresa Casarotto et al., the authors review current knowledge about non-coding RNAs, with miRNAs exclusion, related to human papillomaviruse (HPV)-driven cancers. The manuscript is important in the field and comprehensively describes findings concerning ncRNA role in HPV-driven cancerogenesis. The new version of the manuscript is significantly improved and only minor concerns that should be addressed.
Minor points:
- “NcRNAs as potential diagnostic biomarkers in HPV-driven cancers” subchapter, lines 637-643. The authors write about circRNAs as biomarkers but they do not indicate any specific circRNA neither in this paragraph nor in Table 4
- In table 4, there is “OISI”. In the main text “OIS1”
- lines 191-193: ” Mature miRNAs and siRNAs are transcribed as precursors that undergo a series of splicing events in the cytoplasm, and act in both nucleus and cytoplasm.” A few mistakes in one sentence. Splicing takes place in the nucleus. Splicing is not involved in biogenesis of canonical miRNAs (only mirtrons). The same for siRNA.
Author Response
May 11, 2020
Ref: Revised version of Manuscript # Cancers-771105 Version 2
Dear Reviewer,
We thank you very much for providing further suggestions and comments to our manuscript entitled: "Beyond microRNAs: emerging role of other non-coding RNAs in HPV-driven cancers" by M Casarotto et al.
Following is a point-by-point reply to your comments, together with the changes that have been made in the new version of the manuscript.
- NcRNAs as potential diagnostic biomarkers in HPV-driven cancers” subchapter, lines 637-643. The authors write about circRNAs as biomarkers but they do not indicate any specific circRNA neither in this paragraph nor in Table 4.
Our comment: Upon Reviewer’s suggestion, this paragraph has been removed.
- In table 4, there is “OISI”. In the main text “OIS1”.
Our comment: Thanks to the Reviewer, we have corrected this typographical error.
- Lines 191-193: ” Mature miRNAs and siRNAs are transcribed as precursors that undergo a series of splicing events in the cytoplasm, and act in both nucleus and cytoplasm.” A few mistakes in one sentence. Splicing takes place in the nucleus. Splicing is not involved in biogenesis of canonical miRNAs (only mirtrons). The same for siRNA.
Our comment: We thank the Reviewer for this observation. This sentence has been rephrased to: “Mature miRNAs and siRNAs are transcribed as precursors that undergo a series of nuclear and cytoplasmic processing events, and act in both nucleus and cytoplasm”.
We sincerely hope that you will consider this revised version of the manuscript suitable for publication in Cancers, and I look forward to hearing from you at your earliest convenience.
Kind regards,
Elisabetta Fratta, PhD
Corresponding author